# Deformation Estimation for Time Series InSAR Using Simulated Annealing Algorithm

**DOI:** 10.3390/s19010115

**Published:** 2018-12-31

**Authors:** Wei Duan, Hong Zhang, Chao Wang

**Affiliations:** 1Key Laboratory of Digital Earth Science, Institute of Remote Sensing and Digital Earth, Chinese Academy of Sciences, Beijing 100094, China; duanwei15@mails.ucas.ac.cn; 2University of Chinese Academy of Sciences, Beijing 100049, China

**Keywords:** InSAR, TSInSAR, deformation parameter estimation, simulated annealing

## Abstract

Time series interferometric synthetic aperture radar SAR (TSInSAR) is one of the most important surface deformation monitoring techniques, and has been widely used in geodesy. Deformation estimation is one of the main steps of TSInSAR processing, so an effective and efficient algorithm is necessary. Present algorithms have some limitations such as computing c osts or errors caused by local extremums. In this work, a novel deformation estimation method based on the simulated annealing (SA) algorithm is proposed to handle this problem. The SA algorithm uses a random search to avoid local extremums and thus can be more likely to get the global optimal solution of deformation. By adopting a better annealing method, this algorithm gets high precision deformation results in less time than most present algorithms. In addition, it can estimate complex nonlinear deformation without adding any computing costs. The results, tested on the real SAR data, confirm the reliability and effectiveness of the SA-based deformation estimation algorithm.

## 1. Introduction

In recent years, the interferometric synthetic aperture radar (InSAR) technique [1] has been widely used in monitoring slow and subtle terrain displacements as the phase difference between two radar observations provide a wealth of information about the elevation and displacement of a region. Only considering high coherent pixels, time series InSAR (TSInSAR) solves the problem of temporal and geometrical decorrelations or atmosphere effects in conventional InSAR processing methods. As a result, surface deformation with a high resolution and high precision can be acquired through the analysis of the interferometric phase in a stack of differential interferograms using the TSInSAR method. While deformation estimation is one of the main steps in the processing of TSInSAR, a lot of deformation calculations are inevitable if processing regions with large areas. Hence, there is a need for an accurate, effective, and reliable deformation estimation algorithm.

Currently, the TSInSAR technique has developed into five categories [2]: Permanent scatterer interferometry (PSI) [3,4], small baseline subset (SBAS) [5,6,7], the Permanent scatter and SBAS (PS-SBAS) combined method [8,9,10], SqueeSAR [11] and TomoSAR [12]. Among these methods, SBAS uses the least squares (LS) algorithm to get the deformation parameters from the unwrapped phases, and the estimated results rely on the accuracy of the phase unwrapping of interferograms. PSI estimates the deformation parameters for the arcs of two permanent scatterer (PS) points, and the algorithms used in PSI can deal with both the wrapped and unwrapped phases. PS-SBAS methods, such as Stanford Method for Persistent Scatterers (STAMPS) [13,14] and temporally coherent point InSAR (TCP-InSAR) [15], also rely on phase unwrapping or assume the baselines and arcs are small enough to ensure the phase gradient of two points is immune to phase ambiguities. SqueeSAR deals with the distributed scatterer (DS) and re-estimates the phases for the DS and then uses the same algorithm in PSI to estimate the deformation for both PS and DS. TomoSAR replaces the deformation model with the tomographic imaging model and uses spectral estimation or compressed sensing algorithms [16] to obtain the deformation results. In summary, the deformation estimation methods used in different TSInSAR methods vary greatly. This paper mainly focuses on the deformation estimation methods in PSI.

There are several deformation estimation algorithms used in PSI, including (1) the two-dimensional regression algorithm, which is used in Interferometric Point Target Analysis (IPTA) [17]. It considers two linear relations: Perpendicular baseline and temporal baseline. Linear regression analysis is used in the perpendicular baseline domain and in the temporal domain, and the related slopes correspond to the terrain height and linear deformation. However, the algorithm in IPTA adopted the nonlinear regression method and needed a complex iterative search when the observed phase was wrapped. (2) ILS algorithm is used in the Spatio-Temporal Unwrapping Network (STUN) [18]. It is adopted because the differential phase model can be seen as an integer optimization model. The Least-squares AMBiguity Decorrelation Adjustment (LAMBDA) method was used to solve this problem by using the Z-transform to eliminate the integer ambiguities and then was followed by a sequential least-squares search. The ILS algorithm has the highest probability of integer variable estimation, but its computing cost is high, caused by the computing of the covariance matrix. (3) A two-dimensional solution search algorithm is used in PSInSAR, Persistent Scatterer Pairs InSAR (PSP) [19,20,21] and Quasi-Persistent Scatterer (QPS) [22]. This algorithm avoided the complex modeling by directly searching in the solution space to maximize the object function. This algorithm is simple and flexible, but it cannot take into account accuracy and efficiency, both of which are influenced by the iterative steps. The estimated result may drop into the local extremums if the true deformation and height are not near zero. (4) The Conjugate Gradient Method is used in the coherent pixel technique (CPT) [23] to search the deformation results, and it has the same problem of local extremums as the two-dimensional solution search method.

Recently, some intelligent optimization methods, such as the simulated annealing (SA) algorithm [24], have been widely used for solving the optimization problem. Using a random search based on a given acceptance criterion, the SA algorithm can escape from local extremums. By borrowing the idea of SA, this work presents a novel deformation estimation method which overcomes the limitations of traditional methods. The algorithm is similar to the two-dimensional solution search algorithm while it uses the acceptance criterion to guide the search direction. It takes less time than present algorithms and can get highly accurate results of deformation. It can estimate nonlinear deformation without adding much computing cost. The SA-based deformation estimation algorithm has been tested both on simulated data and on real space satellite data, and the results confirm its effectiveness.

## 2. Deformation Estimation Model

Given a series of *N* co-registered SAR images, according to the standard PSI processing, the first step is to select PS points from SAR images. Then we select a common master image and generate N-1 differential interferograms between other SAR images and the master image; the topographic and flattened phases of the interferograms are removed by using the external digital elevation models (DEM). Then, the differential interferometric phases for the PS points are obtained, and they mainly include the residual topographic phase $\varphi_{topo}$, the deformation phase $\varphi_{def}$, the atmosphere phase $\varphi_{atm}$, the orbit error phase $\varphi_{orbit}$ and the noise phase ε.

Amongst the phases listed, the atmospheric phase $\varphi_{atm}^{k}$ is due to the propagation delay by the atmosphere, while the orbit error phase is mainly caused by inaccurate orbit parameters. Because these phase components are both considered to be correlated in a spatial domain, they can be reduced by the phase differential between two neighbor pixels. In PSI processing, the double difference phase of two pixels is acquired after constructing a PS measurement network (e.g., Delaunay triangular network). The double difference phase of the arcs between pixel i and pixel j for interferogram k is given as follows:
(1)$$\Delta \varphi_{ij}^{k}=\varphi_{ij,topo}^{k}+\varphi_{ij,def}^{k}+\varepsilon_{ij}^{k}$$

For the topography phase, it can be modeled as the linear function of the spatial perpendicular baseline based on the geometry relation of InSAR [25] and is given as follows:(2)$$\varphi_{topo}=-\frac{4\pi }{\lambda }\frac{B_{⊥}}{R\cdot \sin\theta }h=\alpha \cdot h$$where λ is the wavelength of radar carrier signal; R is the slant range between the pixel and radar; θ is the slant angle; $B_{⊥}$ is the spatial perpendicular baseline; h is the relative height (or residual height if external DEM is used in PSI) and α is the phase factor of height.

As for the deformation phase, define the deformation model as the function of the temporal baseline T and some parameters $P=(p_{1},p_{2},\ldots \ldots ,p_{K})$ may be polynomial parameters or seasonal parameters [26,27], i.e., def=d(T;P). Then, the deformation phase is given as follows (define β as the phase factor of deformation):(3)$$\varphi_{def}=-\frac{4\pi }{\lambda }d(T;P)=\beta \cdot d(T;P)$$

Plug Equations (2) and (3) into Equation (1), and it can be written as follows:
(4)$$\Delta \varphi_{ij}^{k}=\alpha_{ij}^{k}\cdot \delta h_{ij}+\beta_{ij}^{k}\cdot \delta d(T_{ij}^{k};P)+\varepsilon_{ij}^{k}$$where $\alpha_{ij}^{k}$ is the spatial phase factor for interferogram; $\beta_{ij}^{k}$ is the temporal phase factor for interferogram; $\delta h_{ij}$ is the relative height between pixel i and j; $\delta d(T_{ij}^{k};P)$ is the relative deformation between pixel i and j; and $\varepsilon_{ij}^{k}$ is the noise phase.

However, the observed phase is wrapped, so Equation (4) contains the extra integer variables. Assuming the phase ambiguity for the phase between pixel i and j is $m_{ij}^{k}$, Equation (4) should be written as follows:(5)$$\Delta \varphi_{ij}^{k}=\alpha_{ij}^{k}\cdot \delta h_{ij}+\beta_{ij}^{k}\cdot \delta d(T_{ij}^{k};P)+\varepsilon_{ij}^{k}+2\pi \cdot m_{ij}^{k},k=1,2,\ldots N-1$$

For *N* interferograms, there will be *N* equations, but more than *N* variables existed due to the phase wrapping. Thus, Equation (5) cannot be easily solved by matrix inversion, and deformation estimation is considered as a nonlinear inversion problem in a two-dimensional regression algorithm or as an integer optimization problem in an ILS algorithm and in the two-dimensional solution search algorithm. Because a two-dimensional solution search algorithm is very simple and finds results in its solution space without using complex models, it has been popular among the PSI methods, such as PSInSAR and PSPInSAR. However, this algorithm is inefficient and affected by the local extremums when the region has very complex deformation fields.

## 3. Methodology

### 3.1. SA Algorithm for Deformation Estimation in TSInSAR

Similar to the two-dimensional solution search method, the SA algorithm avoids the complex integer optimization model by iteratively searching results in the solution space and optimizing the object function. The object function f is usually defined as residual phase temporal coherent coefficient γ.
$$\phi_{k}=\varphi_{k}-\alpha_{k}\cdot h-\beta_{k}\cdot d(T_{k};P),k=1,2,\ldots N$$
(6)$$\gamma =\frac{1}{N}|\sum_{k=1}^{N}e^{j(\varphi_{k}-\phi_{k})}|$$
f=−γ

However, different from the direct solution search, the SA algorithm is a stochastic computational method and utilizes the probabilistic acceptance criterion to guide the search direction and effectively escape from the local extremums. SA algorithm originates in the physical annealing of molten solids. During annealing, SA accepts, not only the improved solution, but also the deteriorated solution, based on the acceptance criterion. At the beginning of the annealing, as the temperature is high, the worse solution can be accepted; with the temperature reducing, only the better worse solution can be accepted; and, at the end of annealing, no worse solution is accepted, and the iterative solution will converge. Because of this feature, the SA algorithm can get out of the local extremums and has more opportunities to find the optimal solution.

Simulated annealing mainly requires three functions to control the iteration process: Cooling schedule, random solution and generator acceptance criterion. To apply the algorithm to the deformation estimation problem, the following functions are used.

(1) Cooling schedule—which defines how the temperature is reduced as the cooling progresses. Geometric cooling is the frequently used schedule and can be given as follows:
(7)$$T_{k}=T_{0}/{\left(\frac{k}{L}\right)}^{q}$$where $T_{0}$ is the initial temperature, $T_{k}$ is the *k*th iteration temperature, L is the iteration length and q is the cooling factor that can adjust the cooling rate to accelerate the iteration process.

(2) Random solution generator is used to generate the new state to find the object function minimum. For the deformation estimation problem, the object function f in Equation (6) is the function of P and h; it means to generate new values of P and h. The SA algorithm usually uses uniform distributed random number as the random solution generator in order to escape from the potential extremums.
$$h=uniform(h_{\min},h_{\max}),$$
(8)$$p_{i}=uniform(p_{i,\min},p_{i,\max}),p_{i}\in P$$where $p_{i}$ and h are the deformation parameters and height, respectively, and $p_{i}$ is in the range of $\left[p_{i,\min},p_{i,\max}\right]$ while h is in the range of $\left[h_{\min},h_{\max}\right]$.

(3) Acceptance criterion is the rule that decides whether a worse solution is accepted, based on a probability function. Usually, the Boltzmann distribution is used as this probability criterion, given as follows:
(9)$$prob=e^{-\frac{\Delta f}{T}}$$

$\Delta f=f^{\prime }-f$, is the object function increment, $f^{\prime }$ is the object function for the new state and T is the temperature of current iteration.

Although the SA algorithm is theoretically guaranteed to converge on the global optimal solution with probability one, there are still some problems in estimating deformation parameters. The above annealing method may have some problems, such as low convergence speed and estimated errors with limited iteration steps. To make the algorithm more efficient and reliable, a better annealing method is needed. Therefore, some changes should be made to the annealing functions, especially for the random solution generator and acceptance criterion. By adopting a good annealing process, this algorithm can obtain the optimum solution and is faster than the two-dimensional solution search algorithm.

### 3.2. A Better Annealing Method

(1) Random solution generator: Using the method of a random number generator like Equation (8), the search process will be time-consuming and have a low-efficiency, especially when the variable range is large. Considering that the variables are consistent, a new solution can be generated around the current solution, and the iterative steps $dp_{i}$ and dh are used. The new random solution generator can be given as follows, taking h as an example:
μ=rand(0,1),
$$dh=\mu^{k}\cdot (h_{\max}-h_{\min})$$
(10)$$h^{\prime }=h+dh$$

Generate a uniformly distributed random number μ in [0,1], and then use the exponential function as the mapping function to transform the random number and get the iterative step dh (k is the iteration record). One thing to note is that the new solution $h^{\prime }$ should be in its range of $\left[h_{\min},h_{\max}\right]$.

Figure 1a shows the exponential function $y=\mu^{k}$ varying with k (k = 1, 4, 200). It can be learned that the iterative step can be very large at the beginning of the iteration, and thus a new solution can jump to any value in the whole solution space. As k (iteration record) increases, the step decreases and the new solutions gradually converge on the final results.

(2) Acceptance criterion: The object function increment Δf in Equation (9) sometimes may be very small and would be better if normalization is used. Then the new acceptance criterion can be given as follows:
(11)$$prob=e^{-\frac{1}{T}(\frac{\Delta f}{\left|f\right|}/\varepsilon )}$$where f is the current object function value and ε is the sensitivity factor (usually $10^{-8}$), which is used to measure the relative increment. To use the acceptance criterion, generate a uniform distribute random number r and calculate prob according to Equation (11). Then compare prob and r: If prob>r, the new solution will be accepted.

The curve of the new acceptance criterion function varying with k is shown in Figure 1b, which has similar characteristics to the Boltzmann distribute function characteristics. At the beginning of the iteration, the probability of accepting a bad deteriorate solution is high to escape from the local extremums; then, as the iteration record k increases, the better deteriorate solution can be accepted with a high probability. Finally, no deteriorated solution is accepted with the probability closed to 0, and this algorithm is the same as a two-dimensional solution search algorithm at this time. In this way, the new function contributes to the convergence.

By using the above annealing method, the efficiency and quality of the proposed algorithm is improved. In addition, the algorithm becomes steady and reliable, which is not influenced by the local extremums. Moreover, the proposed algorithm can get good estimations of the height and deformation with higher precision and less iteration numbers. In general, the SA-based deformation estimation algorithm for TSInSAR is described in Algorithm 1.

**Algorithm 1** SA-based deformation estimation algorithm **Input:** the double difference phase of two pixels with a stack of interferograms.1: Initialize iterative variables: $L,T_{0},P,h$;2: Calculate the object function value f by (6);3: For k=1:L4:   Cool and calculate the temperature $T_{k}$ by (7);5:   generate new solution $P^{\prime }$ and $h^{\prime }$ by (10), then calculate $f^{\prime }$ by (6);6:   calculate the object function increment $\Delta f=f^{\prime }-f$;7:   if Δf < 08:    accept the new solution and update P,h,f;9:   else,10:    calculate the acceptance probability prob by (11);11:    generate a uniform distributed random number r;12:    if prob>r13:     accept the new solution and update P,h,f;14:    end15: end **Output**: the final solution P,h.

## 4. Experiment

### 4.1. Test 1, Simulation Experiment

The first test was a simulated experiment, and the simulated model used the quadratic function $y=ax^{2}+bx+ch+n$ where a and b are the estimated deformation parameters, c is the estimated height, x is the deformation phase factor, h is the height phase factor, n is the noise and y is the phase. We assumed that the interferograms number was N=30 and that the estimated parameters were a=0.389233, b=1.258649 and c=0.826543. n was the Gaussian distributed random noise with its amplitude 0.2 (normalized by the signal amplitude), and y was wrapped to simulate the observed phase. According to Algorithm 1, we used the iterative number L=800, the cooling factor q=2 and the search ranges a∈[−1,1], b∈[−10,10] and c∈[−30,30], and the iterative initial values were all set to be 0. We plotted the object function f varying with deformation parameters a and b as in Figure 2.

From Figure 2a, it can be seen that there are many peaks in the search space. Therefore, many traditional optimization methods will fall into the local extremums and get the wrong estimated results. The two-dimensional solution search algorithm only has the local search ability and will stay at the nearby local extremum, unable to get the optimal result. However, the SA-based algorithm in this work can get a good result.

The search process is illustrated in Figure 2b, and it can be seen that the result is quickly converged to its optimum with a few search numbers. Thus, the SA-based algorithm can get out of the local extremums and acquire the optimal solution. Besides, we can also see that the SA-based algorithm has not only the global search ability at the beginning of the iteration but also the local search ability at the end of the iteration.

The ILS algorithm also gets good results for this model. Here, we compared the SA and ILS algorithm estimated results with different noise levels. We plotted the estimated results distributed with different noise levels as shown in Figure 3.

From Figure 3, we can learn that these two algorithms get similar results with Gaussian noise, and ILS algorithm is slightly better than SA algorithm (ILS has the highest probability of integer variables estimation [28]). Both algorithms satisfy the accuracy need of the deformation estimation problem. As for time complexity, the ILS algorithm is affected by the number of interferograms, and adding more variables will increase its complexity too. However, the SA-based algorithm only relies on its iteration numbers. In this test, the average runtime of the ILS algorithm on MATLAB was 207 ms, but the SA-based algorithm ran in much less time at around 46 ms, nearly two times faster than the ILS algorithm.

### 4.2. Test 2, Real Data Experiment

In this test, we used 17 TerraSAR-X overhead images of the Beijing area in China, taken between 2011 and 2012. The standard PSI was used to process the data: First, we used the amplitude dispersion method to select PS points as shown in Figure 4a; second, we generated 16 differential interferograms and used the spatial filter to smooth the interferograms as shown in Figure 4b; then we constructed the PS measure network and used the SA-based algorithm to estimate the deformation and residual height; finally, we used the MCF phase unwrapping method to unwrap residual phase interferograms (subtracting from the estimated deformation phase and height phase) and got the residual non-modeled deformation phase through atmospheric filtering (temporal high-pass and spatial low-pass filtering). The estimated linear deformation image is shown in Figure 4c.

To better verify the proposed algorithm, we compared the estimated deformation with the leveling data. There were 34 leveling benchmarks measured for the Wenyuhe Bridge as shown in Figure 5a, which was measured twice, in March and September. The second level data were transformed into yearly average linear deformations. Then we extracted the bridge results from the whole region. To ensure that the leveling and PS points have approximately the same reference, the PS point closest to the leveling reference position was selected as the deformation reference. As shown in Figure 5b, the yellow point was the reference.

Subtracting the reference deformation offset for all bridge points, we got the final bridge deformation results and displayed them in Google Earth, as shown in Figure 6.

For all the PS points on the bridge, apart from some error points, the estimated deformation got almost the same results with ILS. The residual non-modeled phase for them is very small and the linear estimated deformation fits the deformation trend well, which can be seen on the curve from Figure 6. It is also seen that the small non-modeled deformation for the bridge appears as in the non-periodical fluctuated form due to a thermal effect. Since the most residual phase standard deviations are small enough, our estimated results are reasonable.

To compare the leveling results with nearby PS points, we plotted both the linear estimated results profile along the bridge in Figure 7. It can be seen that the leveling result (blue) fluctuates greatly, since the vibration of the bridge structure causes many leveling errors. Fitting the leveling result with quadratic curves, there is a distinct deformation increasing trend (black) from the southern end of the bridge to the northern end of the bridge, approximately 18 mm/year. We can see that the linear deformation of the PS (red circle) estimated by the SA-based algorithm fits the trend well. Thus, the effectiveness of the proposed algorithm is well certified.

## 5. Conclusions

Deformation estimation is one of the main steps of TSInSAR processing, and it can be seen as an optimization problem. As the traditional methods have some drawbacks, such as local extremums or computational efficiency, we present a new method based on the simulated annealing algorithm. By using different functions of acceptance criterion and random solution generator, this algorithm can get a good result of deformation with high accuracy and high efficiency. The SA-based deformation estimation algorithm is flexible, reliable and fast. Moreover, it is not influenced by the local extremums. Therefore, this algorithm can be suitable for the deformation calculations in TSInSAR processing.

## Figures and Tables

**Figure 1 sensors-19-00115-f001:**
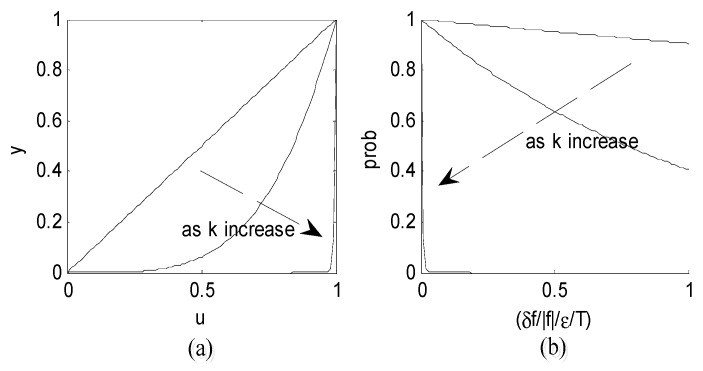
(**a**) The function curves of the random solution generator $y=\mu^{k}$, k = 1, 4, 200 and (**b**) the function curves of the acceptance criterion $prob=e^{-\frac{1}{T}(\frac{\Delta f}{\left|f\right|}/\varepsilon )}$.

**Figure 2 sensors-19-00115-f002:**
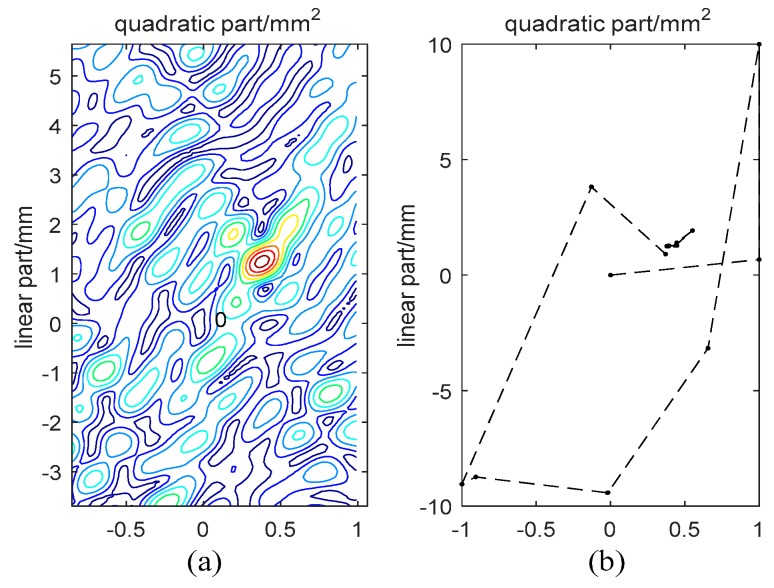
(**a**) The object function f varies with a and b, with the range a∈[−1,1], b∈[−10,10]; and (**b**) the search process for SA-based algorithm.

**Figure 3 sensors-19-00115-f003:**
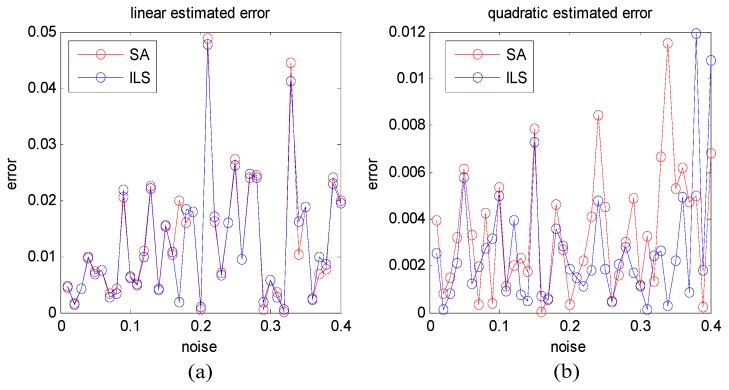
With different Gaussian white noise levels, the linear (**a**) and quadratic (**b**) estimated errors of SA (red) and ILS (blue) algorithm.

**Figure 4 sensors-19-00115-f004:**
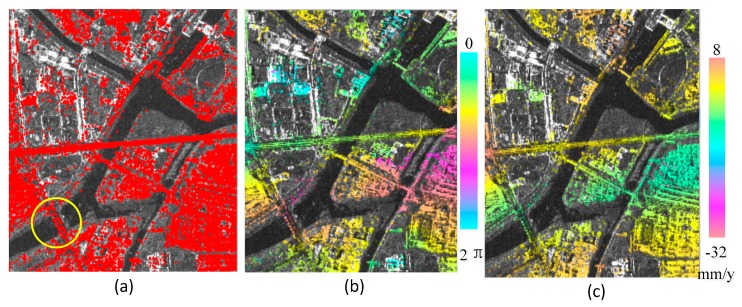
(**a**) Permanent Scatterer (PS) points of the selected region (yellow circle represents Wenyuhe Bridge); (**b**) the filtered differential interferogram of PS points; and (**c**) the linear deformation results of the region.

**Figure 5 sensors-19-00115-f005:**
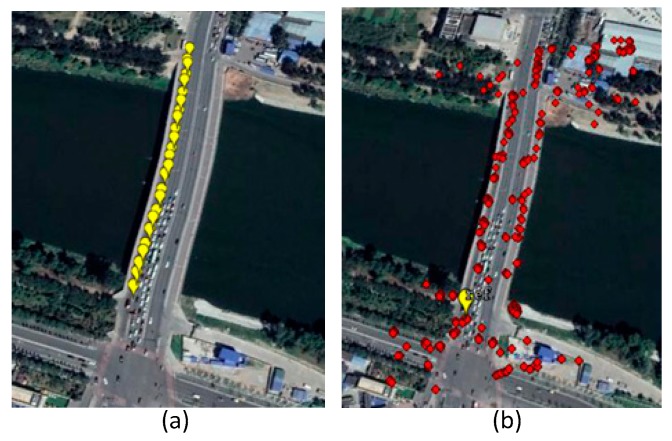
(**a**) The leveling points (yellow dots) and (**b**) the PS points (red dots) and the reference point (the yellow dot) on Google Earth.

**Figure 6 sensors-19-00115-f006:**
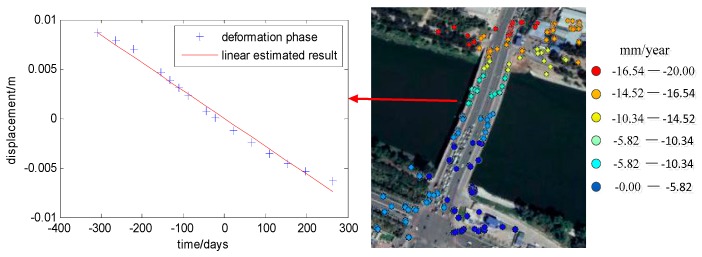
The linear deformation results of the Wenyuhe Bridge on Google Earth.

**Figure 7 sensors-19-00115-f007:**
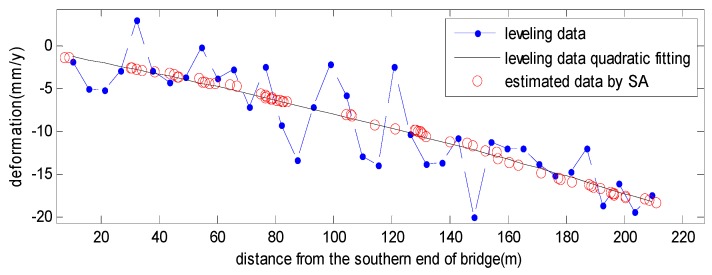
The linear deformation profile along the bridge comparison between leveling (blue dot line), and the estimated result by the SA-based algorithm (red circle).

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
