# Peer review of "Deformation Estimation for Time Series InSAR Using Simulated Annealing Algorithm"

_sensors, 2018, doi:10.3390/s19010115_

Round 1
Reviewer 1 Report
The paper proposes to use SA methods for estimating integer unknowns and retrieve the deformation field for time-series InSAR applications.
The weak point is the relatively scarce novelty of the presented approach. The idea to use SA for solving such a class of problems is not new. The authors should demonstrate, by referring to the literature, that the application of SA in such a context is new.
Also, the introduction to Time-Series InSAR technique is very partial. The authors must refer to the large literature on the field (PS, SBAS, SqueeSAR, hybrid methods, etc ...) and then focus on a particular class or method.
Section 2 must be improved and Section 3, which is the core of the paper, must be revised in the light to make it evident the novelties.
Author Response
Dear the reviewer,
We really appreciate the positive feedback of the two referees and own many thanks to their reviews. We agree with these suggestions and have revised the manuscript accordingly. Below is our response to his/her comments resulting in some clarification. We hope these revisions resolve the problems and uncertainties pointed out by the referee. In the manuscript, the blue color parts are revisions suggested by the referee.
Regards,
Hong Zhang
zhanghong@radi.ac.cn

Reviewer 2 Report
In this manuscript, Simulated Annealing (SA) is used to perform Time Series DInSAR processing. The proposed algorithm is clearly presented and experiments with simulated data show that accuracy is similar to the one of the ILS method, and a better computational efficiency is obtained. My only concern is on the example of application to real data. The employed procedure is very briefly described at the end of page 7 (lines 212-217) and several points are not clear to this reviewer:
- which parameters are estimated by SA optimization? It seems only height and linear displacement coefficient. If so, a simple 2D linear regression would suffice. Perhaps, also the quadratic term is estimated, as in the example with simulated data? Please clarify.
- It seems that SA is used as a preliminary step, followed by phase unwrapping and by retrieval of "nonlinear phase through atmospheric filtering (temporal-spatial filtering)". The latter step is not clear, please better explain what you did. In addition, maybe it is better to unwrap phase before SA optimization: did you try that?
- if you (lines 213-214) "use SA-based algorithm to estimate the deformation and height", you already have deformation, so why subsequent steps are needed? Again, please better explain.
Finally, although the manuscript is fairly well written, further proofreading would be useful. For instance, at the beginning of the Introduction, "...has been used widely used in monitoring..." -> "...has been widely used in monitoring...". In addition, use 10-8 instead of 1e-8.
Author Response

(The authors gave the same response as above.)

Round 2
Reviewer 1 Report
The paper has sufficiently been improved after the revision round.